# Role of Advanced Glycation End-Products and Other Ligands for AGE Receptors in Thyroid Cancer Progression

**DOI:** 10.3390/jcm10184084

**Published:** 2021-09-10

**Authors:** Agnieszka Bronowicka-Szydełko, Łukasz Kotyra, Łukasz Lewandowski, Andrzej Gamian, Irena Kustrzeba-Wójcicka

**Affiliations:** 1Department of Medical Biochemistry, Wroclaw Medical University, Chałubińskiego 10, 50-368 Wrocław, Poland; lukasz.kotyra@umed.wroc.pl (Ł.K.); lukasz.lewandowski@umed.wroc.pl (Ł.L.); irena.kustrzeba-wojcicka@umed.wroc.pl (I.K.-W.); 2Department of Immunology of Infectious Diseases, Hirszfeld Institute of Immunology and Experimental Therapy, Polish Academy of Sciences, Weigla 12, 53-114 Wrocław, Poland; andrzej.gamian@hirszfeld.pl

**Keywords:** AGE, glycation, HMGB1, inflammation, RAGE, S100A4, thyroid cancer

## Abstract

To date, thyroid cancers (TCs) remain a clinical challenge owing to their heterogeneous nature. The etiopathology of TCs is associated not only with genetic mutations or chromosomal rearrangements, but also non-genetic factors, such as oxidative-, nitrosative-, and carbonyl stress-related alterations in tumor environment. These factors, through leading to the activation of intracellular signaling pathways, induce tumor tissue proliferation. Interestingly, the incidence of TCs is often coexistent with various simultaneous mutations. Advanced glycation end-products (AGEs), their precursors and receptors (RAGEs), and other ligands for RAGEs are reported to have significant influence on carcinogenesis and TCs progression, inducing gene mutations, disturbances in histone methylation, and disorders in important carcinogenesis-related pathways, such as PI_3_K/AKT/NF-kB, p21/MEK/MPAK, or JAK/STAT, RAS/ERK/p53, which induce synthesis of interleukins, growth factors, and cytokines, thus influencing metastasis, angiogenesis, and cancer proliferation. Precursors of AGE (such as methylglyoxal (MG)) and selected ligands for RAGEs: AS1004, AS1008, and HMGB1 may, in the future, become potential targets for TCs treatment, as low MG concentration is associated with less aggressive anaplastic thyroid cancer, whereas the administration of anti-RAGE antibodies inhibits the progression of papillary thyroid cancer and anaplastic thyroid cancer. This review is aimed at collecting the information on the role of compounds, engaged in glycation process, in the pathogenesis of TCs. Moreover, the utility of these compounds in the diagnosis and treatment of TCs is thoroughly discussed. Understanding the mechanism of action of these compounds on TCs pathogenesis and progression may potentially be the grounds for the development of new treatment strategies, aiming at quality-of-life improvements.

## 1. Introduction

Thyroid cancer (TC) is the most common endocrine carcinoma, constituting over 95% of endocrine cancers and 3.4% of all carcinomas diagnosed annually all over the world [1]. The incidence of TCs has tripled within last 35 years in western European countries [2]. Moreover, thyroid cancer incidence in the United States between 1974–2013 increased, on average, 3.6% per year [3]. Apart from the increase of TCs incidence, an increase in mortality associated with TCs could be seen, since, during 1994–2013, the incidence-based mortality increased 1.1% per year (95% confidence interval 0.6–1.6%) [3]. These observations may indicate that an evolution of thyroid cancer biology may have an impact on both its incidence rate and its related mortality.

According to the World Health Organization classification, malignant thyroid cancers can be divided into five main histological types [4]: papillary (PTC), follicular (FTC), poorly differentiated (PDTC), medullary (MTC), and anaplastic (ATC). Each of these types, except for MTC (derived from parafollicular, C cells), is derived from follicular cells. Research undoubtedly shows the complexity of thyroid tumors, as some of these tumors may be of mixed (follicular-parafollicular) origin [5]. In addition, many malignant neoplasms of the thyroid gland are lymphomas (most often B-cell) and sarcomas. Figure 1 shows division of malignant TCs and the descriptions of particular types of TCs are shown in Appendix A.

The diagnosis and implementation of proper treatment in case of thyroid disorders, including TCs, is problematic. These disorders are characterized by slow progression of variable, non-schematic symptoms or other indications, including earlier disorders in thyroid function, operations, radiotherapy of thyroid gland, diabetes, malignant anemia, megaloblastic anemia, leukotrichia, hereditary thyroid diseases, and adrenal insufficiency [6]. The implementation of drugs and agents containing lithium or iodine could also induce different metabolic changes. Thyroid cancers express alterations in some genes (a. o. BRAF, RAS, RET, TERT) probably associated with environmental factors [7,8]. Some of these mutations, an example being BRAF^V600E^ (common in PTC), are shown to impair thyroid iodide-metabolizing genes [9]. Although the majority of genetic alterations related to TCs are codified (referring to somatic mutations), the etiology of TCs is still unknown [10]. Ionizing radiation is the only known environmental factor leading to gene mutations, inducing carcinogenesis. Recent scientific reports indicate the possible role of chemicals (phthalates, bisphenols) and heavy metals (cadmium, copper, and lead) in etiopathology of disorders of the thyroid gland [11]. The stromal changes in thyroid neoplasms may be associated with alterations in fibroblasts, endothelial cells, tumor-associated inflammatory cells, and numerous calcifications, known as ‘Psammom bodies’.

Although thyroid nodules are detected in 50% of healthy patients, only 5–15% of them are malignant [12]. Nevertheless, developing new differentiation and treatment strategies for TCs is of high importance, as its incidence has the highest impact on mortality among all of the endocrine cancers. The introduction of different screening methods, such as ultrasonography and fine-needle aspiration (FNA), contributed to the lowering of the number of TC-associated thyroid operations. However, cytological assessment most often does not enable quick, preoperational diagnosis of each type of TC, since approximately 20–25% of biopsy results are classified as indeterminate (according to the Bethesda system), requiring surgical procedures before providing any further histological assessment [13]. Although only approximately 4% of thyroid nodules are carcinomas and require surgery, 15–30% of thyroid cytology results are not conclusive [14], possessing diagnostic and prognostic dilemmas [15]. This fact led to an uprising interest in thyroid cell oncogenesis [15,16] and the need for the discovery of many potentially promising blood-circulating biomarker candidates, most of them being thyroid-specific transcripts of thyroglobulin [17,18,19,20,21,22,23,24], thyroid stimulating hormone receptor [25,26], thyroid peroxidase [27,28], or sodium/iodide symporter [29,30]. However, the discrepancy between the studies into the usefulness of these mRNAs in screening for TCs (mainly in case of thyroglobulin mRNA) led to an increasing demand to come up with better prognostic markers.

Many studies aimed to describe the prognostic value of advanced glycation end-products (AGEs) in the context of TCs. The idea to analyze these compounds stems from the fact that the increased risk of developing TCs is primarily associated with chronic hyperglycemia [31] which, through triggering lipid peroxidation, chronic inflammation, oxidative DNA modifications, and loss/gain of protein function (many of which have an antioxidative potential) [32,33,34], induces a release of carbonyl compounds and reactive oxygen species (ROS), promoting the process of glycation (thus—formation of AGEs). These glycation-derived compounds are characterized by slow elimination from the bloodstream, as virtually no enzymes were shown to catalyze reactions of their decomposition. Glycation of nucleic acids is associated with neoplastic changes, glycation of enzymes and lipoproteins, and alterations in metabolic pathways, while glycation of hormones disturbs signaling pathways, leading to abnormal transcriptional activity, aberrations in immune response, and oxidative stress intensification [35]. Protein glycation, along with protein aggregation, increases the risk of transformation of physiological cells into neoplastic due to changes in the function of proteins involved in cell metabolism. Both these processes are interrelated, as protein modifications which trigger protein aggregation may be induced by AGEs [36]. Moreover, AGE-associated modifications in the extracellular matrix promote its remodulation and/or dysfunction, increasing the risk of TCs. By binding to matrix proteins, AGEs cause an increase in matrix stiffness, which is shown to be correlated with tumor progression [37] and associated with alterations in response to tumor treatment [38] and activation of the receptor for glycation end-products (RAGE)—further intensifying inflammatory processes (and thus also promoting carcinogenesis). The mentioned changes in matrix stiffness due to glycation and/or oxidative damage may also be affected by genetic mutations/polymorphisms which also promote matrix remodulation. One of the most common (frequency of 29–83% [39,40,41]) mutations associated with PTCs, the BRAF^V600E^, increases the matrix stiffness due to phosphorylation of MEK1/2 and ERK1/2. This occurrence does not only promote PTC progression, but also triggers tumor cell migration and increases PTC aggressiveness. A knockdown of BRAF^V600E^, however, may alleviate this effect [42]. As mentioned before, this mutation also affects the thyroid iodide-metabolizing genes such as NIS, coding for sodium/iodide symporter. Interestingly, the effects of BRAFV600E have been ascribed to TGFβ mechanism of action. This information may be very vital in personalized anti-PTC treatment as it has been shown that targeting nitric oxide synthase 4 (one of TGFβ key effectors) with siRNA decreases BRAFV600E-induced repression of NIS [9]. The general outcome of PTCs may also be associated with aberrant methylation of genes engaged in TC progression. As shown by Kikuchi et al. [43], gene methylation in PTC may be correlated a priori with BRAF/RAS oncogene. Interestingly in thyroid glands, the aforementioned causative factor of glycation-chronic hyperglycemia is further promoted by oxidative damage, since malignant thyrocytes, especially poorly differentiated, were shown to overexpress insulin receptor [44]. Moreover, in TCs, as in breast cancer, the accumulation of AGEs and RAGE activation is connected with an increase in estrogen receptor expression on cell surface, having an effect on genesis, reprogramming, and progression of TCs [45].

The complex topic of the influence of AGEs on the development of TCs demands updating and thorough discussion, as the opportunity to understand this association has arisen lately with the results of new studies being published. This review covers the influence of RAGE, AGEs, AGE precursors, and ligands for RAGE on the progression of different TCs and discusses the use of these compounds in diagnostic and therapeutic strategies.

## 2. Advanced Glycation End-Products and Their Possible Linkage with Thyroid Cancers

### 2.1. The Glycation Process, Advanced Glycation End-Products, and Their Receptors

Glycation occurs spontaneously in a cascade of various non-enzymatic reactions (dehydration, oxidation, condensation, isomerization, cyclization) between amino groups of proteins, lipids, nucleic acids, and reducing carbohydrates or alpha-oxoaldehydes, leading to the formation of advanced glycation end-products. Glycation may affect the structure and function of many proteins and peptides, including enzymes (including enzymatic antioxidants) [46], hormones (including the thyroid hormone) [47], lipoproteins [48] and other transport-related proteins [49], signal transducers [50], and antibodies [51,52]. AGEs were first described 90 years ago [53]. However, many of them are still unknown [54].

Although glycation is known to be coexistent with physiological processes associated with aging [55], it is also intensified in metabolic disorders such as chronic hyperglycemia [56,57,58]. Many chronic diseases including cancer [59,60,61,62], neurodegenerative disorders [63,64], stroke [65], rheumatic [66,67] and cardiovascular [68,69] diseases and diabetes [70], with their implications, may occur through cellular dysfunction and AGE-induced inflammation [71]. AGEs increase the synthesis of reactive oxygen species (ROS), escalating oxidative stress and increasing the probability of the occurrence of apoptosis (through protein modifications and lipid peroxidation of polyunsaturated fatty acids in cellular membranes) [72], at the same time suppressing antioxidative mechanisms by inactivation of enzymatic antioxidants and alterations in antioxidative potential of non-enzymatic antioxidants (mainly, glutathione) [73,74]. The glycation process may also have beneficial effects in the body, as AGEs have been shown to inhibit osteogenic differentiation of adipose-derived stem cells by activating DNA methylation and inhibiting the Wnt/β-catenin pathway in vitro. Induction of DNA methylation allows regeneration of ASC bone tissue in diabetic osteoporosis [75].

Due to AGEs’ rather long half-time and association with oxidative or carbonyl stress, peroxidation of lipids, and formation of advanced lipoxidation end-products [76], AGEs are often considered as auxiliary markers in numerous diseases, proving to be of prognostic or even therapeutic value [77].

The receptors for AGEs are situated on the surface of cellular membranes. Interestingly, these receptors are involved in AGE-associated homeostasis, as some of them may not only exacerbate inflammation and oxidative stress, but also control the number of AGEs, preventing their accumulation. The most-studied AGEs receptors include [78]: RAGE, AGE-R1 (oligosaccharyl transferase-48, OST-48), AGE-R2 (80K-H phosphoprotein), AGE-R3 (galectin-3), SR-A (macrophage scavenger receptor A), SR-BI and SR-BII (two classes of scavenger receptor B, CD36), ERMs: ezrin, radixin, moesin [79], LOX-1 (lectin-like oxidized LDL receptor-1), and sRAGE (soluble RAGE) [80].

The most-studied AGEs receptor, RAGE, belongs to the immunoglobulin superfamily [81]. RAGE is found on the surface of various cells and tissues, including pulmonary alveoli [82], monocytes/macrophages, endothelial, and dendritic cells [83]. It has been shown that RAGE does not only recognize AGEs (endogenous or food-derived), but also other compounds, e.g., advanced protein oxidation products (engaged in oxidative stress), fibrillar β-sheet forming amyloids characteristic of Alzheimer’s disease [84], S100 protein family members (Ca^2+^-binding modulators), and high mobility group box-1 (expressed in cancer) (Figure 2).

AGE-RAGE interaction evokes constant and strong cellular response, inducing an intracellular cascade of inflammatory reactions which lead to release of proinflammatory cytokines [85], intensification of oxidative stress, increased risk of coagulopathies [86], overexpression of different extracellular matrix proteins (such as collagen or laminin) [87], and activation of secondary signal transmitters, e.g., kinase C. These processes promote alterations in cellular stimulation, migration, and proliferation, leading to pathological condition such as diabetic complications, sclerosis, Alzheimer’s disease, inflammation, and tumor development [88]. Most of these conditions are due to RAGE-associated release of intermediates of various cell signaling pathways, such as ROS, p21ras, erk1/2 (p44/p42), MAPK kinase [89] p38, SAPK/JNK MAP kinases, rhoGTPases, phosphoinositol-3 kinase, and intermediates of JAK/STAT pathway.

AGEs are considered to be potential markers of oxidative stress in several inflammatory and autoimmune diseases, including autoimmune thyroid disease [90]. Although little information is available on the role of this interaction in thyroid autoimmunity, it has been observed that AGEs concentration was elevated in Hashimoto’s thyroiditis (HT) [91].

### 2.2. AGEs, RAGE, and Possible Therapeutic Application of RAGE Antagonists in the State of Carcinogenesis

As mentioned before, especially in state of their massive accumulation, AGEs and other compounds (β-amyloid peptides, high mobility group box 1, some proteins of S100 family, β-sheet fibrils, prions and NFκβ) bind to RAGE, inducing changes within cell physiology via several pathways, including Ras-extracellular signal-regulating kinase 1/2 [92], p38 mitogen-activating protein kinase [93], JAK 1/2 [94], NADPH oxidase [95], and Cdc 42/Rac [96]. RAGE, upon being triggered by these compounds, triggers the activation of activator protein-1 [97], several other signal transducers and activators of transcription (STAT) [98], and, most importantly—NFκβ, being one of the key targets for RAGE signaling [99]. The NFκβ-associated, upregulated expression of cell adhesion molecules, growth factors (including TGFβ) and proinflammatory cytokines (IL-6, TNFα), along with activation of NADPH oxidase and glycation of proteins of antioxidative capacity, promote the lack of pro-/antioxidative homeostasis and accumulation of ROS and reactive nitrogen species (RNS) [100]. Therefore, a process of transient RAGE activation may shift into chronic state of inflammation due to RAGE activation through gradual increase in ROS/RNS, lesions within protein structure (and function), and positive feedback loop due to the ability of NFκβ to bind to RAGE.

Owing to these facts and the well-studied association of (meta-)inflammation with odds of tumor incidence [101], AGEs and RAGE have been (and are, still) studied as potential markers of carcinogenesis [102,103]. Increased expression of RAGE, induced by its ligands, was found to promote cancer metastasis; moreover, prevention from accumulation of AGEs and/or activation or RAGE inhibited metastasis of various tumors, including lung, breast, laryngeal, prostate, hepatocellular carcinomas, and melanoma, inter alia [104,105,106,107]. Targeting RAGE with RNAi or its modified, antagonistic ligands was shown to inhibit tumor growth and invasion [108,109,110]. Moreover, as RAGE promotes angiogenesis (via affecting the expression of VEGF and other factors), silencing the signal derived from RAGE activation was associated with inhibition of angiogenesis induced by colorectal carcinoma, both in vitro and in vivo [111,112,113]. Interestingly, some of the widely-applied drugs might be able to affect RAGE activation, since statins were shown to be able to downregulate RAGE, inducing decrease in VEGF [114].

## 3. AGEs as a Possible Link between Oxidative Stress and the Incidence of TCs

### 3.1. Insights on Carcinogenesis in Context of Glycation, Inflammation, Oxidative/Carbonyl/Nitrosative Stress, and Their Impact on Cell Integrity and Function

The hypothesis regarding the formation of TCs is based on the existence of small subpopulations of epigenetically and genetically transformed stem cells which may form cancer cells of different phenotype. According to this way of understanding the pathogenesis of TCs, anaplastic thyroid carcinoma cells (ATC) are derived from human thyroid cancer stem cells [115], formed under condition of simultaneous genetic/epigenetic transformations within proper, physiological thyroid stem cell line. Non-transforming thyroid stem cells, however, may generate multipotential progenitor cells (thyroblasts), which may easily transform into TCs (PTC or FTC). Benign tumors are formed out of thyrocyte precursor cells (prothyrocytes) [116]. Such intracarcenous heterogeneity of thyroid carcinomas (TCs) develops as a result of genetic and epigenetic changes in tumor cells, which most frequently is a process induced by interactions between genetic and non-genetic factors or under the influence of alterations within tumor microenvironment (mainly: inflammatory cells, fibroblasts, and extracellular matrix) [117]. Chronic hypoxia developed within tumor microenvironment may activate hypoxia inducible factor 1 (HIF-1) and NF-κB, inducing proliferation and stimulation of angiogenesis, inhibition of apoptosis, and activation of regulatory proteins associated with various cascades such as: MAPK, PI_3_K, tyrosine kinases, ion channels, and receptors connected to protein G. RAGE, similarly to TLRs and P2X7, is activated by hypoxia factor HIF-1α which also induces activation of NF-κB within activated B cells and expression of genes coding for proinflammatory cytokines [118,119]. Prevalent glycolytic metabolism and increased pro-oxidative potential within cancer cells induce accumulation of AGEs, virtually promoting shifts in interleukin profile within tumor microenvironment [120]. Moreover, as opposed to common rationale, that glycation occurs mainly intracellularly, it has been shown that TCs, regardless of whether benign or malignant, are associated with intense glycation, although the microenvironment of TCs was more prone to this process, compared to tumor cells [121]. As dicarbonyl compounds (precursors of AGEs), RAGE receptors and their ligands: HMGB1, K-Ras, S100A4, and S100A8 are engaged in the initiation and progression TCs, these compounds are being investigated as possible therapeutic aims (Figure 3).

Genetic variability within single tumor is caused by many factors, including telomerase activation (this enables aging cells to overcome telomere crisis), genetic mutations in genes involved in different processes or appearance of propulsive mutation being essential for cancer initiation [122]. The non-genetic heterogeneity is mainly connected with epigenetic abnormalities which lead to genome instability, with aberrant methylation of promoter regions of one or many genes responsible for regulation of cell cycle (including post translational modifications of histones), and abnormalities within microRNA expression are examples of such aberrations. Interestingly, the formation of TCs may also be promoted by excessive thyroid gland stimulation by pituitary thyroid gland stimulating hormone (TSH). The physiological processes of proliferation and replication of DNA are preserved within cells of various tissues in the state of physiological concentrations of ROS and RNS. However, since dicarbonyl compounds (main products of: lipid peroxidation, anaerobic glycolysis and protein degradation) such as glyoxal, MG, and 3-deoxyglucosone may directly react with proteins, lipids, and nucleic acids (modifying their structure and influencing tissue microenvironment [123]), these two processes are aberrant due to the state of multifactorial disturbance of pro-/antioxidative homeostasis, associated with inflammation, inactivation of enzymatic antioxidants, modifications of various proteins (including hormones and factors involved in cell differentiation), lipid peroxidation (promoting apoptosis due to modifications within cell membranes), and DNA modifications [124]. Interestingly, in the state of TCs incidence, the antioxidative potential may be able to adapt to glycation, as fluorescent AGEs were positively correlated with glutathione content [125].

As glycation is co-existent with carbonyl/oxidative/nitrosative stress, it is involved in the process of TCs formation. AGEs are able to damage DNA, increasing the risk of TCs’ progression and promoting cancer cell proliferation, migration, and resistance to apoptosis [126]. One of the frequent causes of genetic changes in nuclear DNA is the condensation of nucleic acids as a result of their interaction with carbonyl groups of various compounds (most commonly, aldehydes, and ketones), the concentration of which increases in the state of carbonyl stress induced by lipid peroxidation or glycation. Nucleic acids are characterized by their long half-time, which promotes DNA glycation [127]. The exposure of DNA molecules to methylglyoxal (MG) may lead to DNA damage, delay of meiosis, karyokinetic spindle aberrations, or delay of anaphase I [128]. Furthermore, the following increase in oxidative stress and the accompanying carbonyl stress induce telomere damage, affecting the replication lifetime of the cells. Induced oncogenes activate intracellular signaling pathways affecting the proliferation of thyroid cells, e.g., mitogen activated kinase pathway (MAPK) and extracellular pathways, e.g., extracellular signal regulated kinase pathway (ERK) [129], lipid 3-phosphatidylinositol kinase (PI_3_K) cascade, serine-threonine protein kinase Akt (PI_3_K-Akt), and β-catenin activating cascade [130].

Intracellular transcriptional activity depends on the packing of chromatin by histone proteins. Although TCs have mainly been associated with aberrant histone acetylation/deacetylation [131], histones are also susceptible to glycation due to their long half-time (4–5 months), high content of Lys/Arg, and presence of disordered, nucleophilic tails, which are prone to modifications. Histone glycation induced by presence glucose, ribose, fructose, 3-deoxyglucosone, ADP-ribose, and MG is feasible, as these small molecules (<5 kDa) can easily diffuse through the nuclear membrane into the cell cytoplasm [132]. Apart from glycation, MG also causes histone oxidation due to accumulation of reactive carbonyls. These modifications alter the stability of nucleosomes and chromatin architecture, promoting gene mutations. Intense histone glycation has already been associated with incidence of some types of tumors, e.g., breast cancer [133]. Modified histones show variable cooperative binding with DNA and may induce production of antibodies which are present in lung cancer, prostate cancer, breast cancer, and cancer of head and neck. Further research into histone glycation and immunogenicity of such modified histones may contribute to understanding the role of glycated nuclear proteins in different types of cancer, with thyroid cancers being an example [134].

The carcinogenesis-associated alterations within cell cycle demand cellular adhesion to extracellular matrix, which is granted by ERK kinase signaling and induction of cyclin D1 [135]. It has been shown that the interaction between AGEs and RAGE strengthens the synthesis of signaling mediators such as: ERK, PI_3_K, RAC, and cyclin D [136]. It is important to note that these mediators respond (with activation) to an increase in rigidness of the extracellular matrix. Most malignant tissues (TCs, also) are characterized by stronger rigidity, associated with altered collagen cross-linkage. In the state of increased oxidative stress and co-existing glycation, the number or collagen cross-links increases, increasing extracellular matrix rigidness and activating mediators such as ERK, PI_3_K, and RAC, aiming at strengthening cyclin D1 expression [137].

### 3.2. Selected Advanced Glycation End-Product Precursors and RAGE Ligands as Factors of Thyroid Tumor Progression. Anti-Glycative Treatment as a Potentially-Promising Tool in Future Thyroid Cancer Therapy

#### 3.2.1. Methylglyoxal

One of the most-studied proglycative agents, MG, is more reactive than glucose [138], forming different AGEs as a result of its reaction with proteins, inducing dicarbonyl stress and leading to cell damage [139]. MG-derived dicarbonyl adducts exert complex, pleiotropic effects on healthy and pathological intracellular processes since they change the activity and stability of proteins and, at the same time, induce ROS, enhancing oxidative stress [140]. Changes induced by MG may exert distant time effects [141]. Accumulation of MG-derived hydroimidazole (MG-H1) or argpyrimidine (AP) leads to DNA aberrations, promoting apoptosis [142] and impairment of cell survival function maintenance [143]. Defining the role of MG in ATC incidence/progression and identifying molecules which could decrease the aggressiveness of ATC is still under investigation. Successful therapeutic intervention in ATC so far has been of little probability due to poor knowledge of molecular etiology of this cancer, as ATC is a highly aggressive malignant cancer with features of undifferentiation, and remains resilient to conventional therapy [144]. A research conducted in vitro on aggressive anaplastic thyroid cancer (ATC) cells proved that there is a correlation between aggressiveness of this cancer and intensity of MG-induced dicarbonyl stress [145]. Interestingly, overexpression of Glo1 in some cancer cells (including ATC) allows utilization of the activity of this enzyme in anti-cancer treatment [146]. MG has also been suggested as a potential marker of tumor progression, even though is known for exerting different (even opposing) biological effects [147]. Elevation of MG concentration in ATC is coexistent with a decrease in activity of glutathione-glyoxalase 1 (Glo-1). The role of MG and Glo1 seems to be dependent on cancer type, since cancer tissues and cell lines of different molecular background and MG detoxication rate react differently under stress caused by MG [148]. Another in vitro study confirmed that the invasive/migration properties of model ATC cells were associated with accumulation of MG adducts and implementation of aminoguanidine, and that resveratrol decreased the amount of MG adducts. Moreover, the concentration of MG adducts was negatively associated with the activity of Glo1 and seemed to induce the intensification of ATC in patients [149]. Literature suggests that, in case of ATC, the implementation of MG scavengers (e.g., aminoguanidine) and Glo1 activators (e.g., resveratrol) decreases MG level, activates Glo1 and, as a consequence, eases off tumor aggression due to changes in processes of its invasion and migration. Wang et al. demonstrated feasibility of complete resection, decreased need for tracheostomy, high pathologic response rates, and durable locoregional control with symptom amelioration in patients with BRAF^V600E^–mutated ATC patients with locoregionally advanced disease [150]. Dicarbonyl stress blockade performed by MG scavenger receptors and Glo1 activators are a potential therapeutic strategy in ATC treatment.

#### 3.2.2. High-Mobility Group Box 1 Protein (HMGB1)

HMGB1, a ligand for RAGE and one of the late proinflammatory cytokines, plays a role in the pathogenesis of ATC and PTC, by stimulating monocytes and macrophages and via TLR–mediated secretion of: growth, chemotactic, and angiogenetic factors, associated with inflammation [151]. Mice exposed to ablation of both alleles associated with HMGB were shown to be more prone to oncogenesis due to oversensitization towards K-Ras protein [152]. Mutations activating the RAS pathway in thyroid epithelial cells were directly associated with early and frequent transformations and proliferations of thyroid cancer (follicular thyroid cancer and follicular variant of papillary thyroid carcinoma) [153,154]. Thus, it seems likely that the interaction between K–ras and activated RAGE promotes carcinogenesis in thyroid cells. This assumption is supported by the fact that HMGB1-RAGE interaction promotes ATC and PTC carcinogenesis due to increased expression of miRNA uncoding molecules (associated with tumor proliferation, apoptosis, invasion, and metastasis). The fact that PTC has been shown to be associated with overexpression of oncogenic miRNAs (miR–143–3p, miR–146b, miR-221, miR–222) [155] may be of potential use in future therapy (Figure 4).

The decrease in miR–143–3p expression may be connected with the aggressiveness of PTC [156]. Overexpression of miR–221 occurs in PTC cells and in their neighboring tissues, suggesting that it may virtually be a phenomenon characteristic for early stages of carcinogenesis, as well as developed carcinogenesis [157]. It has been confirmed that miR–146b, miR–221, and miR–222 are overexpressed in PTC [158]. Anti-RAGE antibodies, due to preventing from overstimulation of RAGE, may prove potentially effective in downregulation of miR221 and miR222 expression, which may inhibit ATC and PTC carcinogenesis (Figure 5).

It has been shown that anti-RAGE antibodies successfully blocked HMBG1/RAGE/miR221/222 signaling pathway. Impaired HMBG1–RAGE interaction decreased the expression of miR–221 and miR–222 [159].

#### 3.2.3. S100 Proteins

Several proteins of the S100 family have also been featured in studies into TCs incidence. A representative of this family, the S100A4, plays a key role in proliferation and metastasis of TC cells. S100A4 is localized both in nucleus and cytoplasm, although it may occur extracellularly. It regulates angiogenesis and cell survival and increases invasiveness and metastatic potential of cancer cells [160]. Moreover, in the extracellular fluid, S100A4 interacts with compounds which are localized on cell surface, namely RAGE, annexin II and proteoglycans of heparan sulfate, which are involved in migration of TC cells. A study conducted on FTC, PTC, and ATC cell lines/tissues in which RAGE was stimulated with S100A4 showed that the increase of cellular migration induced by RAGE was dependent on diaphanous 1 (Dia-1)–an intracellular signaling compound which accompanies RAGE, evoking activation of small GTPases: Cdc42 and RhoA. Interestingly, although extracellular S100A4 constantly activated ERK signaling in TC cells, it has been shown that this signaling was not transmitted via RAGE [161]. Therefore, it has been shown that S100A4–RAGE/Dia1–associated signaling may promote TC cell migration. Therapy targeted at the interaction between RAGE, Dia1, and small GTPases may potentially reduce local invasion and metastases in thyroid cancer [162].

Another S100 representative, the S100A8, has been shown to stimulate proliferation of the ATC cells, leading to activation of p38, ERK, and JNK pathways [163]. In studies carried out both in vitro and in vivo on ATC cells, an increased level of S100A8 was detected [164]. Inhibition of S100A8 was considered to be an appropriate therapeutic target, allowing the limitation of cancer cell proliferation and metastasis [165].

## 4. Conclusions

Thyroid cancers (TCs), including papillary (PTC), follicular (FTC), poorly differentiated (PDTC), medullary (MTC), anaplastic (ATC), their mixes, and other rarely appearing TCs, form complicated tumors. These cancers pose a diagnostic and therapeutic challenge.

AGEs, precursors of AGEs, RAGE, and RAGE ligands not being AGEs are all engaged in TCs carcinogenesis, playing different roles during its various stages. Inflammation–related processes such as oxidative, carbonyl, and nitrosative stress promote protein (enzymes, histones, and other structural proteins, such as collagen), DNA, and lipid glycation, which leads to structural changes, altering the biological function of these compounds and promoting mutations of genes associated with the incidence of TCs. The interaction of one of AGEs precursors, the methylglyoxal, with DNA, may further promote mutations and cell cycle disorders.

It is important to note that the deleterious impact of glycation on cell physiology stems not only from various RAGE-associated interactions within thyroid cells, but also from these interactions taking part within the cell microenvironment: AGE–RAGE interaction enhances the synthesis of ERK, PI_3_K, RAC, and cyclin D, which disrupts the cell cycle. HMGB1 promotes the incidence of ATC and PTC through increasing the expression of uncoding molecules of miRNA, which play a role in cell proliferation, apoptosis, invasion, and metastasis. S100A4 is engaged in proliferation and metastasis of thyroid cancers (follicular, papillary, and anaplastic) because it promotes tumor cell migration. Moreover, the RAGE/Dia1/small GTPases interaction may successfully reduce local invasion and metastasis. S100A8–RAGE interaction activates p38, ERK, and JNK pathways in tumor cells.

RAGE, its ligands, and compounds engaged in glycation may be investigated as potential aims of TC treatment, as some of them have been shown to affect thyroid cancer aggressiveness. The increase in methylglyoxal concentration, often observed in ATC, may be lowered by treatment with aminoguanidine and resveratrol. Anti–RAGE antibodies, however, were shown to effectively decrease the concentration of miR221 and miR222 (increased in thyroid cancer). Moreover, the inhibition of S100A8 proglycative function of S100A8 was proposed to be an appropriate therapeutic target in patients with anaplastic thyroid cancer.

Although, in the context of thyroid cancer, the amount of evidence against glycation and its related processes is scarce, it shows that the impact of glycation on different aspects of thyroid cell life and death, albeit complex and demanding laborious trials of success and error, may evolve into clinically relevant strategies for thyroid cancer treatment, aiming to control the outcome of tumor disease and the related alterations in the quality of life.

## Figures and Tables

**Figure 1 jcm-10-04084-f001:**
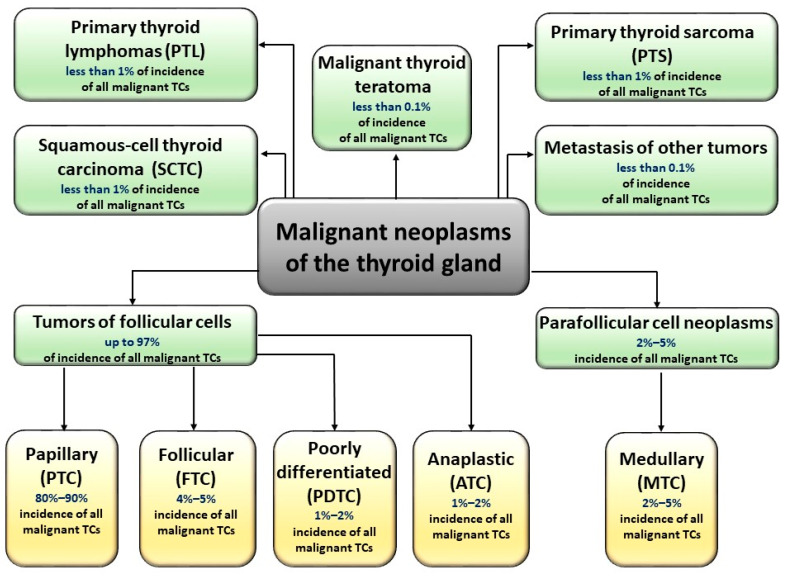
Types of malignant thyroid carcinomas. Malignant neoplasms of the thyroid gland are derived from parafollicular C cells (medullary tumor cancer, MTC) or from follicular cells (papillary (PTC), follicular (FTC), poorly differentiated (PDTC), and anaplastic (ATC). Some TC tumors are of mixed (follicular-parafollicular) origin. TCs could also occur in the form of lymphomas (most often B-cell), sarcomas, teratomas, and squamous-cell thyroid carcinomas.

**Figure 2 jcm-10-04084-f002:**
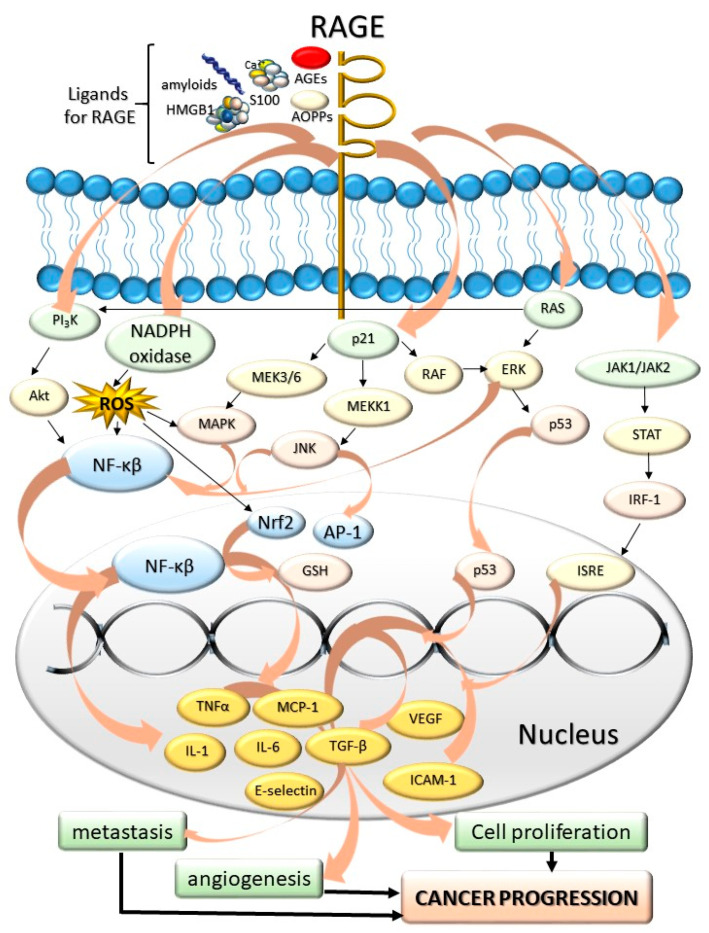
The influence of AGEs and other ligands for RAGE: amyloids, high mobility group box-1 proteins (HMGB), advanced protein oxidation products (AOPPs), S100 proteins on the cancer progression. The AGE-RAGE interaction induces various cascades associated with cell division and tumor formation. The mediators of these cascades include JAK/STAT chain; p21, PI_3_K kinase, regulators of signal transduction (Ras proteins), and NADPH oxidase. As a result of RAGE activation, numerous compounds are expressed within the cell nucleus and secreted into the bloodstream. These compounds include proinflammatory cytokines, interleukins and necrosis factors (tumor necrosis factor α, TNF-α, IL-1, IL-6), different extracellular matrix proteins (collagen or laminin), secondary signal transmitters, regulators of proliferation, differentiation, mitosis, cell survival, and apoptosis (Mitogen-activated protein kinase, MAPK, nuclear factor kappa-light-chain-enhancer of activated B cells, NF-κβ). As these compounds are accumulated, the cells are under the effect of intensified proliferation, angiogenesis, and metastasis-promoting cancer progression. Abbreviations: RAGE—Role of Advanced Glycation End-Products, AGEs—Advanced glycation end-products, PI_3_K—phosphoinositide 3-kinase, AKT—protein kinase B, NADPH—nicotinamide adenine dinucleotide phosphate, MAPK—mitogen activated protein kinase, MEKK1—mitogen-activated kinase kinase 1, RAF—rapidly accelerated fibrosarcoma, p53—tumor protein P53, STAT—signal transducer and activator of transcription, IRF-1—interferon regulatory factor 1, JAK—Janus kinases, ISRE—interferon stimulated response elements, RAS—regulators of signal transduction, ERK—extracellular signal-regulated kinases, ROS—reactive oxygen species, JNK—c-Jun N-terminal kinase, AP-1—activator protein 1, Nrf2—nuclear respiratory factor 2, GSH—glutathione, NF-κβ—nuclear factor kappa-light-chain-enhancer of activated B cells, TGF-α—tumor growth factor alpha, MCP-1—monocyte chemoattractant protein-1, VEGF—vascular endothelial growth factor, ICAM-1—intercellular adhesion molecule 1, TGF-β—tumor growth factor beta, IL-1—interleukin 1, IL-6—interleukin 6.

**Figure 3 jcm-10-04084-f003:**
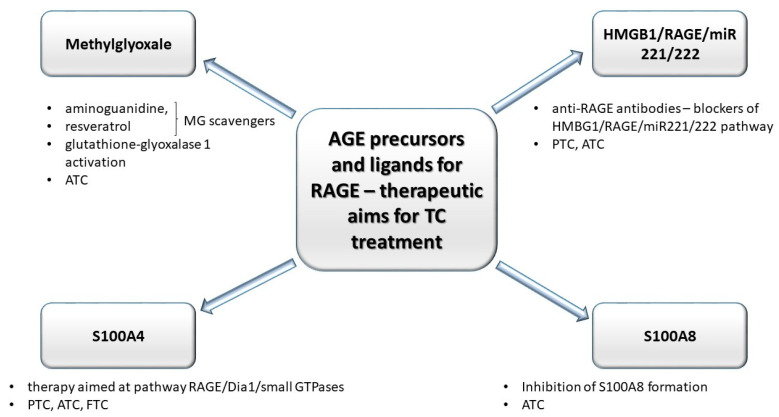
Compounds engaged in the glycation: precursor of AGE (methylglyoxale, MG) and some ligands (high mobility group box–1 proteins (HMGB1), S100 proteins) for AGE’s receptors (RAGEs) may be utilized as therapeutic aims in anti-TCs treatment strategies. MG scavengers such as aminoguanidine and resveratrol decrease the level of MG adducts. Anti-RAGE antibodies block HMGB1/RAGE/miR221/222 pathway (engaged in progression of anaplastic tumor cancer (ATC) and papillary tumor cancer (PTC)). The decrease in S100A8 formation inhibits ATC progression. Moreover, in anaplastic tumor cancer (ATC), the MG concentration may be controlled by altered activity of glutathione-glyoxalase 1 (Glo-1). S100A4–RAGE/Dia1-associated signaling pathway may promote migration of TC cells.

**Figure 4 jcm-10-04084-f004:**
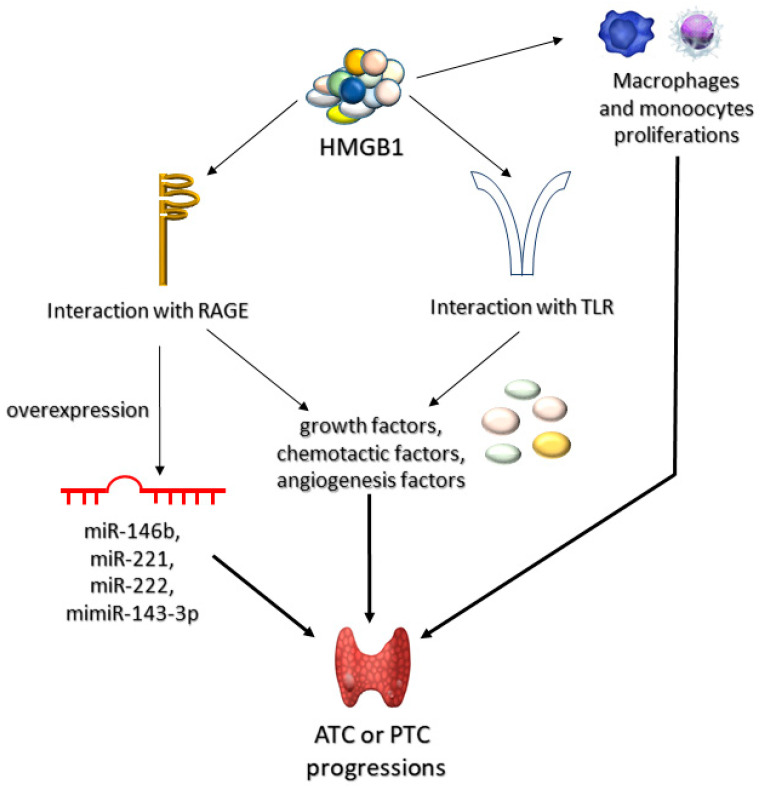
Role of HMGB1 in progression of anaplastic (ATC) and papillary (PTC) thyroid tumors. High mobility group box-1 proteins (HMGB1) can lead to ATC or PTC progression through three mechanisms of action. Firstly, HMGB1 can induce proliferation of macrophages and monocytes. Secondly, it may interact with AGEs receptors (RAGEs) or toll-like receptors (TLRs). The interaction HMGB1 with RAGEs can lead to either overexpression of microRNA (miR-146b, miR-221, miR-222, or mimiR-143-3p) or to synthesis of various growth, chemotactic, or angiogenetic factors. Accumulation of these compounds leads to progression of ATC or PTC.

**Figure 5 jcm-10-04084-f005:**
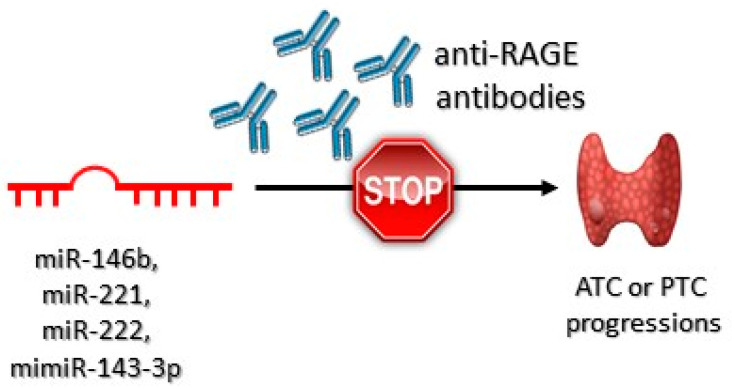
Role of the anti-RAGE antibodies in anaplastic (ATC) and papillary (PTC) thyroid tumor progression. These antibodies bind to RAGE receptors preventing HMGB1-RAGEs interactions. As a consequence, ATC or PTC progression is inhibited.

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
