# Peer review of "Role of Advanced Glycation End-Products and Other Ligands for AGE Receptors in Thyroid Cancer Progression"

_jcm, 2021, doi:10.3390/jcm10184084_

Round 1

Reviewer 1 Report

The study addresses an interesting and current problem about the role of advanced glycation end-products and other ligands for AGE receptors in thyroid cancer progression. However, I have certain objections related to the paper. The major criticism regards the conclusion that are too short and describe poorly the sense of this review. I suggest rewriting the conclusion in order to enrich the review. Below you can find some minor points to review.

Minor points: 

-    Line 52 “The two latter…” please explicit the type of cancer.
-    Line 61-66 I suggest to break the sentence.
-    Line 125 and line 149  remove “Brief information”.
-    Line 165 I think the reference is not correct in the sentence, please move it. 
-    Line 223 I suggest to remove “Brief”
-    Line 234 please check the brackets.
-    Line 323-325 I suggest to change the title of this section
-    Line 342-368 some references were used several times (for example: 104, 137,141 and ). However, you used more than one times the same references for several topics in the manuscript. I think you could find new references instead to use the same.
-    I suggest to write some lines o explain the figures and not only the title. Moreover, you should report the abbreviation used in the figure.

Author Response

  1. The major criticism regards the conclusion that are too short and describe poorly the sense of this review. I suggest rewriting the conclusion in order to enrich the review. Below you can find some minor points to review.

Answer: Thank You for this comment and Your suggestion. The conclusion has been revised in order to enrich the review.

Thyroid cancers (TCs): papillary (PTC), follicular (FTC), poorly differentiated (PDTC), medullary (MTC), anaplastic (ATC), their mixes and others rarely appearing TC’s cancers form complicated tumors. These cancers pose a diagnostic and therapeutic challenge.

AGEs, precursors of AGEs, RAGE and RAGE ligands not being AGEs are all engaged in TCs carcinogenesis, playing different roles during its various stages. Inflammation-related processes: oxidative, carbonyl and nitrosative stress, promote protein (enzymes, histones and other structural proteins, such as collagen), DNA and lipid glycation, which lead to: structural changes, altering the biological function of these compounds and promoting mutations of genes associated with the incidence of TCs. The interaction of one of AGEs precursors, the methylglyoxal, with DNA, may further promote mutations and cell cycle disorders.

It is important to note that the deleterious impact of glycation on cell physiology stems not only from various RAGE-associated interactions within thyroid cells, but also from these interactions taking part within cell microenvironment: AGE-RAGE interaction enhances the synthesis of ERK, PI3K, RAC and cyclin D, which disrupt the cell cycle. HMGB1 promotes the incidence of ATC and PTC through increasing the expression of uncoding molecules of miRNA, which play their role in: cell proliferation, apoptosis, invasion and metastasis. S100A4 is engaged in proliferation and metastasis of thyroid cancers: follicular, papillary and anaplastic because it promotes tumor cell migration. Moreover, the RAGE/Dia1/small GTPases interaction may successfully reduce local invasion and metastasis. S100A8-RAGE interaction, activates p38, ERK and JNK pathways in tumor cells.

RAGE, its ligands and compounds engaged in glycation may be investigated as potential aims of TC treatment, as some of them have been shown to affect thyroid cancer aggressiveness. The increase in methylglyoxal concentration, often observed in ATC, may be lowered by treatment with aminoguanidine and resveratrol. Anti-RAGE antibodies, however, were shown to effectively decrease the concentration of miR221 and miR222 (increased in thyroid cancer). Moreover, the inhibition of S100A8 proglycative function of S100A8 was proposed to be an appropriate therapeutic target in patients with anaplastic thyroid cancer.

Although, in the context of thyroid cancer, the amount of evidence against glycation and its related processes is scarce, it shows that the impact of glycation on different aspects of thyroid cell life and death, albeit complex and demanding laborious trials of success and error, may evolve into clinically-relevant strategies for thyroid cancer treatment, aiming to control the outcome of tumor disease and the related alterations in the quality of life.

  1. Line 52 “The two latter…” please explicit the type of cancer.

Answer: Thank You for this suggestion. This fragment: “Interestingly, TCs differ from one another in pathogenesis and tumor structure, malignancy level, prognosis and treatment strategy. Moreover, the two latter are also de-pendent on clinical stage, which could be shown with an example of PTC, showing high chance or positive outcome after treatment by thyroid gland removal and iodine radiation – in its early stages [5], or associated with poor prognosis and high chance of recurrence or post-treatment metastases – in its late stages [6]” has been removed.

  1. Line 61-66 I suggest to break the sentence.

Answer: Thank You for Your comment. This fragment has been broken into three sentences:

The diagnosis and implementation of proper treatment in case of thyroid disorders, including TCs, is problematic. These disorders are characterized by slow progression of variable, non-schematic symptoms or other indications, including: earlier disorders in thyroid function, operations, radiotherapy of thyroid gland, diabetes, malignant anemia, megaloblastic anemia, leukotrichia, hereditary thyroid diseases, adrenal insufficiency [6]. The implementation of drugs and agents containing lithium or iodine could also induce different metabolic changes.

  1. Line 125 and line 149 remove “Brief information”.

Answer: Thank You for Your comments. “Brief information on the glycation process” has been changed into “The glycation process” and “Brief information on advanced glycation end-products and their receptors” has been changed into “Advanced glycation end-products and their receptors”.

  1. Line 165 I think the reference is not correct in the sentence, please move it.

Answer: Thank You for pointing it out. We apologize for these editorial errors. The proper reference is:

Goldin, A.; Beckman, J.A.; Schmidt, A.M.; Creager, M.A. Advanced glycation end products: sparking the development of diabetic vascular injury. Circulation. 2006, 114, 597-605, doi: 10.1161/CIRCULATIONAHA.106.621854.

  1. Line 223 I suggest to remove “Brief”

Thank You for Your comments. The subtitle “Brief insights on carcinogenesis in context of glycation, inflammation, oxidative/carbonyl/nitrosative stress and their impact on cell integrity and function” has been changed into “Insights on carcinogenesis in context of glycation, inflammation, oxidative/carbonyl/nitrosative stress and their impact on cell integrity and function”.

  1. Line 234 please check the brackets.

Answer: Thank You for pointing it out. We apologize for the editorial error. The parenthesis has been closed.

Benign tumors are formed out of thyrocyte precursor cells (prothyrocytes).

  1. Line 323-325 I suggest to change the title of this section

Answer: Thank You for pointing it out. The title of this section has been changed into:

Selected advanced glycation end-product precursors and RAGE ligands as factors of thyroid tumor progression. Antiglycative treatment as a potentially-promising tool in future thyroid cancer therapy?

  1. Line 342-368 some references were used several times (for example: 104, 137,141 and ). However, you used more than one times the same references for several topics in the manuscript. I think you could find new references instead to use the same.

Answer: Thank You for this comment. Bibliography has been re-checked, new literature positions have been added to the paper. Each of the reference has been used only for one topic. Added literature positions have been marked in yellow color.

  1. I suggest to write some lines of explain the figures and not only the title. Moreover, you should report the abbreviation used in the figure.

Answer: Thank You for this comment. We have added the explanation of the figures and reported the abbreviation used in the figures.

Reviewer 2 Report

Interesting review providing new insights about pathogenesis, progression and suitable future treatment approaches to thyroid cancer.

I find the paper well-written and surely conceptualized by researchers experienced in the field (as I can also see from the pubmed profile especially of the first author).

  1. Introduction
  • When addressing epidemiology of thyroid cancer, please refer to the SEER 2017 study by Lim et al., where an increase not only of incidence, but also of mortality (classified as incidence based) was defined. This indicates an evolution of thyroid cancer biology, which impacts on both incidence and mortality.
  • Concerning epidemiology, it seems that some rare thyroid malignancies (metastases, lymphomas, teratomas…) have the same “epidemiological weight” of the follicular-cell derived forms. Authors should transmit to readers that the scenario of thyroid cancer is dominated by differentiated thyroid cancer and specifically by papillary thyroid cancer (the latter account for 85-90% of thyroid malignancies!). In such view, figure 1, despite having nothing wrong, let the reader imagine that all tumors have the same epidemiology and this is a wrong message
  • Interestingly, TCs differ from one another in pathogenesis and tumor structure, malignacy level, prognosis and treatment strategy. Moreover, the two latter are also dependent on clinical stage, which could be shown with an example of PTC, showing high chance or positive outcome after treatment by thyroid gland removal and iodine radiation – in its early stages [5], or associated with poor prognosis and high chance of recurrence or post-treatment metastases in its late stages [6]

Confusing, very confusing. Please reformulate or remove

  • TCs are  caused by genetic and/or environmental factors, often associated with changes within tumor microenvironment [8,9].

It is necessary to be careful about that point. The vast majority of genetic alterations related to thyroid cancer are codified (referring to somatic mutations) (please cite Marotta V, Sciammarella C, Colao A, Faggiano A. Application of molecular biology of differentiated thyroid cancer for clinical prognostication. Endocr Relat Cancer. 2016 Nov;23(11):R499-R515. doi: 10.1530/ERC-16-0372. Epub 2016 Aug 30. PMID: 27578827.). However, mutations are pathogenetic events, etiology is another thing. Etiology of thyroid cancer is widely unknown as the only recognized factor is represented by ionizing radiations. In recent years, a possible etiologic role is emerging for environmental chemicals with thyroid interfering function. Please cite, at this regard, the most recent and updated review about this issue: Marotta V, Malandrino P, Russo M, Panariello I, Ionna F, Chiofalo MG, Pezzullo L. Fathoming the link between anthropogenic chemical contamination and thyroid cancer. Crit Rev Oncol Hematol. 2020 Jun;150:102950. doi: 10.1016/j.critrevonc.2020.102950. Epub 2020 Apr 10.

  • Regarding the issue of nodules with indeterminate cytology, authors should remove the sentences: These unneeded surgeries stem from the fact that, although FNA is of great positive and  negative predictive values in case of distinction of malignant papillary thyroid tumors does not allow differentiation between benign and malignant follicular lesions.

Please state only that about 15-30% of cytology result are not conclusive….indeed many inconclusive cytological findings occur in case of the PTC follicular variant, which is not a follicular lesion

  • When talking about the role of molecular profiling in the diagnostics of thyroid nodules with indeterminate cytology, please cite the following papers: Guerra A, Di Stasi V, Zeppa P, Faggiano A, Marotta V, Vitale M. BRAF(V600E) assessment by pyrosequencing in fine needle aspirates of thyroid nodules with concurrent Hashimoto's thyroiditis is a reliable assay. Endocrine. 2014 Mar;45(2):249-55. doi: 10.1007/s12020-013-9994-y. Epub 2013 Jun 18. PMID:23775008; Marotta V, Bifulco M, Vitale M. Significance of RAS Mutations in Thyroid Benign Nodules and Non-Medullary Thyroid Cancer. Cancers (Basel). 2021 Jul 27;13(15):3785. doi: 10.3390/cancers13153785. PMID: 34359686; PMCID: PMC8345070.
  • In general, please state that the articles refers to oncogenesis of the thyroid cell, so to follicular-derived thyroid malignancies.
  1. Main body
  • I suggest to shorten section 2 into one single paragraph
  • I found the molecular section well articolated and clear for readers. I just ask to include some evidence linking glycation products and the genetic alterations related to thyroid cancer (differentiated), such as BRAFV600E and RAS point mutations, which are the historical ones. Particularly, PTC (the vast majority of differentiated thyroid cancer and therefore of thyroid malignancies) are divided into the clinic-molecular types BRAF-like and RAS-like formed (based on the most recent NGS analyses). Are there some difference about the differential role of glycation products in such subtypes of PTC? Or at least authors may provide some speculation?

Round 2

Reviewer 1 Report

I appreciated your corrections.

This manuscript is a resubmission of an earlier submission. The following is a list of the peer review reports and author responses from that submission.

Round 1

Reviewer 1 Report

The manuscript “Role of advanced glycation end-products (AGEs) and other ligands for AGE receptors (RAGE) in thyroid cancers progression” by Agnieszka Bronowicka-Szydełko et al. treats a very interesting topic about the cause responsible for the thyroid cancer progression. However, I have certain suggestions related to the paper. First of all, reading the paper, I have the impression that either it is some kind of patchwork.

Line 34: please remove tubercles and rewrite the sentence (Although thyroid nodules may be detected in almost 50% of healthy patients, at the age of 60 only 5 – 15% of them are usually malignant).

Line 37 (again) use thyroid nodules instead of “tubercles”.

Lines 36-43: rewrite the sentence: too confused and no data necessary (1970?)

Lines 44-52: please add some references.

Lines 56-62: the sentence: “Diagnosis of thyroid disease and implementation of proper treatment in some disorders of this gland are problematic. This stems from the fact that disorders of thyroid physiology are characterized by slow dynamic of symptoms’ increase and symptoms are often variable. The medical indications for thyroid gland diagnostics are: a history of disorders in thyroid function, surgery, radiotherapy of thyroid gland, diabetes, albinism, malignant anaemia, megaloblastic anaemia, leukotrichia, implementation of drugs and agents containing lithium or iodine.” The same sentence is completely repeated some lines later (lines 71-77).

I am wondering if the authors wrote and submitted the manuscript without read it…

The manuscript cannot be accepted in the present form. I suggest you rewrite some parts, in particular the introduction, to give a coherent guideline to the readers.

Moreover, it is required an extensive review of the English, for the introduction that must be completely rewritten.

Reviewer 2 Report

The authors are attempting in this paper to write a review about the role of advanced glycation end-products and their receptors in thyroid cancer.  However, they have failed to deliver a clear, concise, and understandable discussion of the subject due to the many issues outlined below.  There are far too many corrections needed in this paper for me to specifically list them all so only a very few examples of the many issues that must be addressed will be given.

General comments-

There are many 1) grammatical and spelling errors and unclear, misleading or 2) incorrect use of words, apparent even in the abstract.  More concerning, the writing is unfocussed, disorganized, 3) often repetitive, and while in some cases, though the material is factually correct, its importance is 4) overly emphasized.  Most troubling, occasionally, the material is 5) just plain wrong. 

The authors must get an English-speaking scientist with a background in thyroid cancer biology to correct the grammar, help with better word usage, and correct facts.

Repetitive text needs to be removed and the document also needs to be structured much better. A logical outline may help.

Finally, there absolutely needs to be much more of a focus on how exactly AGE/RAGE biology is specifically affecting thyroid cancer, rather than so much general discussion about the connection of AGE and RAGE to cancer and about thyroid cancer unrelated to AGE/RAGE biology. 

Specific comments-

1) An instance of incorrect grammar and spelling, and unclear use of words can be found in the sentence starting on line 17 of the abstract, “They may induce genes mutations, disturb histone methylation and both initiate and disorder important pathways of cancerogenesis, such as PI3K/AKT/NF-kB, p21/MEK/MPAK, JAK/STAT, RAS/ERK/p53, that induce synthesis of inteleukines, growth factors and cytokines, compounds influencing on metastasis, angiogenesis and cancer proliferation.”.  The grammatically incorrect phrase “They may induce genes mutations...” should be something like “They may induce genes to mutate...”.  The word “interleukins” is misspelled as “inteleukines”, a mistake easy to catch with a spell checker.  The grammatically incorrect and unclear phrase “...compounds influencing on metastasis...” perhaps means “…which are all compounds that influence metastasis…”. 

As another example of unclear use of words, in the sentence starting on line 25, “The aim of this review was to collect information about the role of compounds engaged in glycation in the pathogenesis of TC and to determine their utility in the diagnosis and treatment of TC.”, are they suggesting that compounds engaged in glycation might be useful as thyroid cancer treatments, which is what this sentence says, or do they really mean that an understanding of the mechanism of how glycated compounds affects thyroid cancer may lead to successful TC treatments?

2) For an example of frequent poor choice of words, the use of “tubercles” when speaking of the thyroid (sentence starting at line 34 “Although tubercles of thyroid gland may be detected in 50% of healthy patients at the age of 60 only 5 – 15% 35 thyroid nodules are malignant [3].”) is not the best choice.  Although the term “tubercle” can be used to indicate a cancerous structure, the thyroid has several tubercles that are normal structures e.g. Zuckerkandl tubercles, and the superior and inferior tubercles of the thyroid cartilage.  The use of the word “nodules” to indicate a possible cancerous structure is clearer.

3) Two examples of repetitive text follow.

The sentence starting on line 59, “The medical indications for thyroid gland diagnostics are: a history of disorders in thyroid function, surgery, radiotherapy of thyroid gland, diabetes, albinism, malignant anaemia, megaloblastic anaemia, leukotrichia, implementation of drugs and agents containing lithium or iodine.” is largely repetitive with the sentence starting on line 74 “Medical indications for thyroid gland diagnostics are as follows: earlier disorders in thyroid function, operations, radiotherapy of thyroid gland, diabetes, albinism, malignant anaemia, megaloblastic anaemia, leukotrichia, implementation of drugs and agents containing lithium or iodine, hereditary thyroid diseases, original adrenal insufficiency [8].”.

The information in the phrase “by hereditary factors that means chromosomal changes (gene rearragments: RET/PTC or PAX8/PPARg)” (line 211-212) is later repeated in the sentence close below starting at line 242, “Among the hereditary factors are rearrangmenets in chromosomes with genes encoding tyrosine kinase RET or box transcription factor PAX8 paired with nuclear receptor of transcription factors (PPARγ).”

4) An instance of exaggerated importance can be seen in the sentence starting on line 210 containing the phrase mentioned above, “TC can be caused by genetic factors (mutations), by hereditary factors that means chromosomal changes (gene rearragments: RET/PTC or PAX8/PPARg), by environmental factors (e.g. iodine deficiency in food) and by excessive thyroid gland stimulation by pituitary thyroid gland stimulating hormone (TSH).” Iodine deficiency may lead to a slight increase in the incidence of thyroid cancer, though actually it appears to act as a promoter if anything rather than an initiator of TC (Acta Endocrinol (Buchar). 2018 Oct-Dec; 14(4): 525–526), but if so, its incidence does not compare to the vastly larger proportion of thyroid cancers arising from alterations in the DNA sequence.

5) Finally, several examples showing incorrect facts follow.

There is no mention of albinism or thyroid in the reference 8 listed in example 3) (in sentence starting on line 74 “Medical indications for thyroid gland diagnostics are as follows: earlier disorders in thyroid function, operations, radiotherapy of thyroid gland, diabetes, albinism, malignant anaemia, megaloblastic anaemia, leukotrichia, implementation of drugs and agents containing lithium or iodine, hereditary thyroid diseases, original adrenal insufficiency [8].”). Is this the correct reference?  I cannot find any evidence that thyroid cancer has any connection to albinism.  Where did this “fact” come from? 

The sentence starting on Line 316 states “Successful therapeutic intervention in ATC so far has been of little probability due to poor knowledge of molecular etiology of this cancer – ATC is a highly aggressive malignant cancer with features of undifferentiation that remains resilient to conventional therapy [82]”. This is incorrect.  Please see Thyroid 2019 Aug;29(8):1036-1043 for evidence of a successful therapeutic intervention for ATC based on a good knowledge of the cancer’s molecular etiology.

These examples demonstrate only a few of the many instances that must be addressed, along with a logical reorganization of the material and narrowed focus on the AGE/RAGE-thyroid cancer intersection, before the review might be considered acceptable.